# Origin, genetic diversity and evolution of Andaman local duck, a native duck germplasm of an insular region of India

Arun Kumar De[1]*, Sneha Sawhney[1], Debasis Bhattacharya[1], T. Sujatha[1], Jai Sunder[1], Perumal Ponraj[1], S. K. Ravi[1], Samiran Mondal[2], Dhruba Malakar[3], A. Kundu[1]

1 Animal Science Division, ICAR-Central Island Agricultural Research Institute, Port Blair, Andaman and Nicobar Islands, India, 2 Department of Veterinary Pathology, West Bengal University of Animal and Fishery Sciences, Kolkata, West Bengal, India, 3 Animal Biotechnology Centre, National Dairy Research Institute, Karnal, Haryana, India

* biotech.cari@gmail.com

**Data Availability Statement:** All relevant data are within the manuscript and its Supporting information files.

## Abstract

Domestic ducks are of paramount importance as a cheap source of protein in rural India. Andaman local duck (ALD) is an indigenous avian genetic resource of Andaman and Nicobar islands (ANI) and is mainly distributed in Middle and Northern parts of these islands. Negligence has brought this breed on the edge of extinction necessitating immediate conservation efforts. Here, we report the genetic diversity, population structure and matrilineal genetic root of ALD. Partial mtDNA D-loop sequences were analyzed in 71 ALD samples and analysis revealed 19 polymorphic sites and 13 haplotypes. Estimated haplotype (Hd ± SD) and nucleotide diversity (π ± SD) were 0.881 ± 0.017 and 0.00897 ± 0.00078 respectively. The high genetic diversity of ALD indicates introgression of genetic material from other local duck breeds. In addition, it can be postulated that ALD bearing high genetic diversity has strong ability to adapt to environmental changes and can withstand impending climate change. Phylogenetic and network analysis indicate that ALD falls under Eurasian clade of mallard and ALD forms three clusters; one cluster is phylogenetically close to Southeast Asian countries, one close to Southern part of mainland India and the third one forms an independent cluster. Therefore, ALD might have migrated either from Southeast Asian countries which enjoy a close cultural bondage with ANI from time immemorial or from Southern part of India. The independent cluster may have evolved locally in these islands and natural selection pressure imposed by environmental conditions might be the driving force for evaluation of these duck haplotypes; which mimics Darwin's theory of natural selection. The results of the study will be beneficial for formulating future breeding programme and conservation strategy towards sustainable development of the duck breed.

## Introduction

Duck rearing is a lucrative enterprise in Asia particularly in Southeast Asia and domestic ducks are generally reared for their meat, eggs and feathers and is a crucial food source in the

**Funding:** The work was funded by National Bank for Agriculture and Rural Development (NABARD, Port Blair) and Department of Biotechnology, Ministry of Science and Technology, Government of India (grant no. BT/BI/04/066/2004).

**Competing interests:** No competing interests exist.

rural parts of Southeast Asia [1, 2]. The process of duck domestication has extensively been studied and currently two hypothesis exists; one suggests that wild mallard (*Anas boschas)* is the ancestor of common domestic ducks [3, 4], whereas, the second claims that the domestic ducks are the results of a hybridization event between the mallard (*Anas platyrhynchos*) and the Eastern spot-billed duck (*Anas zonorhyncha*) [5]. Recently, Zhang et al. [6] unveiled the genetic basis of duck domestication through whole genome sequencing based analysis and duck domestication was found to be a complex event and overlapped with the domestication event of other vertebrates [6]. Since domestication, domestic ducks have been spread to various countries and regions with varied climatic conditions and a wide range of duck breeds, varieties and strains have evolved in the process of domestication due to varied selection pressure and breeding strategy. As per 2016 data, the global population of duck (*Anas* spp.) was 1,24 trillion; in which a major chunk (89%) was in Asia [7]. In India, duck raising is very popular as they are very prolific and adaptable to free range system of rearing [8]. Moreover, free-range ducks act as natural predators for insects and snails in water body and paddy fields. As per 20th Livestock Census (2019), Govt. of India, the total backyard poultry in the country is 317.07 million in which domestic ducks account for 3% share (www.vikaspedia.in) and mostly scattered in Eastern, Northeastern and Southern part. In India, there is only one registered duck breed (www.nbagr.res.in), besides several location specific native breeds with huge genetic potential. Negligence has brought some lesser-known breeds to the brink of extinction and merits immediate conservation efforts. Conservation of genetic diversity is the core of sustaining a breed [9]. Characterization including delineation of genetic diversity, population structure, phylogeograpgy, evolution and domestication process of the duck genetic resources is of paramount importance to fully explore the potential of the breeds and boost their conservation.

Several molecular markers like RAPD, RFLP and microsatellites are being used for molecular characterization of animal genetic resources [10–12]. In the last decade, mitochondrial DNA (mtDNA) has emerged as a very powerful and reliable marker in animal population and evolutionary biology [13–16]. The mtDNA, due to its several characteristics including high mutation rate, lack of recombination and maternal inheritance, is extensively used in phylogenetic studies. The control region of mtDNA, the most variable region of mtDNA and having 5–10 fold higher mutation frequency than other parts, is an ideal genetic marker [17, 18] and has been used to understand the genetic makeup, population structure and phylogeograpgy of chicken in Bangladesh [19], duck population of China [20, 21], Indonesia [22], Bangladesh [23], Vietnam [23], Korea [24, 25] and Nigeria [26]. It was also used to deduce the interspecific hybridization of wild mallards with its closely related species in Russia, North Asia, the Aleutian island and mainland Alaska [27, 28]. In India, report on genetic characterization of domestic ducks is very limited. Microsatellite based analysis of four Indian duck population exhibited significant diversity among the population [29].

Andaman and Nicobar islands (ANI), an archipelago located in the juncture of Bay of Bengal and the Andaman sea, are very rich in floral and faunal biodiversity and are the home of several animal genetic resources. Andaman local duck is a native duck breed of ANI, mainly found in Middle and Northern parts of Andaman and contributes significantly to the livelihood and nutritional security of the rural farmers. They are medium in size and attains adult weight of 1.0–1.5 kg. Plumage color is grey-brown to blackish brown, head color is lustrous black green, bill color is orange, feet color is bright orange and tail color is blackish green or brown [30]. Of recent, negligence and introduction of exotic breeds make this duck breed very fragile to extinction and immediate characterization and conservation is on top priority. Formulation of sustainable conservation strategy necessitates information within and between population genetic diversity. Moreover, genetic root and migration pattern of these ducks are

still unknown. Therefore, the present study was aimed to assess the genetic diversity, population structure at mtDNA level and unveil the genetic root and migration pattern of Andaman local duck.

## Materials and methods

### Ethics approval

Ethical permission was granted from the Institute Animal Ethical Committee (IAEC) of ICAR-Central Island Agricultural Research Institute, Port Blair, Andaman and Nicobar Islands, India. Relevant standard guidelines and regulations were followed throughout the study.

### Sampling and DNA isolation

Blood samples from 71 Andaman local ducks kept at farmers' field in North and Middle Andaman were collected. During sampling, due care was taken to avoid genetically related samples based on detailed interviews with the owners. Not more than three randomly chosen samples per village were considered as per FAO guidelines [31]. Information on sampling location is provided in S1 Table. From each bird, approximately 3 ml of blood was drawn from the wing vein using a 23 gauge needle fitted to a 5 mil syringe into a vacutainer containing EDTA. Genomic DNA was isolated using a commercial kit (GSure Blood DNA Mini Kit, GCC Biotech India Pvt. Ltd, Kolkata, India, Cat. No. G4626), following the manufacturer's instructions. The quality and concentration of isolated DNA samples were checked using a BioSpectrometer (Eppendorf, Hamburg, Germany) and DNA samples were then stored at -20˚C until further use.

### PCR amplification and sequencing of mtDNA control region (D-loop)

A fragment of duck mtDNA D-loop (710 bp) was amplified using primers described earlier [23]. PCR was done in a 25 μl reaction volume containing 2.5 μl 10X $Taq$ buffer with 1.5 mM MgCl$_2$, 0.2 mM dNTPs, 1 μM each forward and reverse primer, 1 IU $Taq$ DNA polymerase (GCC Biotech India Pvt. Ltd, Kolkata, India) and approximately 50 ng genomic DNA. The PCR reactions were carried out in a Thermocycler (Eppendorf, Hamburg, Germany) with the cycling conditions as mentioned by Wu et al. [23]. Amplified products were purified and sequenced in both directions by Sanger dideoxy fingerprinting. The generated sequences were edited using Sequencher v 5.4.6 (Gene Codes Corporation, USA).

### Bioinformatics analysis

Mallard reference sequence and representative D-loop sequences of domestic ducks and mallards from different countries representing Europe, Asia and North America were retrieved from GenBank (www.ncbi.nlm.nih.gov) and a summary of the information has been depicted in S2 Table. Mitochondrial control region haplotypes of domestic ducks and mallards are indistinguishable and phylogenetically they cluster together [32, 33]; therefore, the dataset is a admixture of domestic ducks and mallards. Alignment of the sequences was done by ClustalW [34] in MEGAX [35]. Polymorphism and population diversity parameters like number of polymorphic sites, haplotype number and diversity, nucleotide diversity and DNA divergence between populations were computed by DnaSp v 6 [36]. Maximum Composite Likelihood model [37] implemented in MEGAX was used to compute pair-wise genetic distance among different sequences. To determine the evolutionary relationship among different sequences, median-joining networks were constructed in Network v 10 [38] or PopART ver. 1.7 [39] with

default settings. Phylogenetic relationship among mitochondrial sequences was inferred using the Neighbor-Joining method [40] with the Tamura-Nei model [41] as implemented in MEGAX following 1,000 bootstrap replications. Bayesian phylogenetic analysis of the sequences was established using MCMC model in BEAST v1.10.4 [42]. For phylogenetic tree construction, we trimmed extra nucleotides from our sequences and GenBank retrieved sequences to make a homogeneous length of 475 bp. Population differentiation was calculated by Wright's F-statistics [43] and assessment of genetic variance among and within the population was done by analyses of molecular variance (AMOVA) [44] in Arlequin v 3.5 [45] with 16,000 permutations. To delineate population expansion, mismatch distribution was calculated in DnaSp v 6 [36] and neutrality tests (Tajima's D test, Fu's FS test, Fu and Li's D test, Fu and Li's F test) were carried out in DnaSp v 6 [36] and Arlequin v 3.5 [45].

## Results

### Polymorphisms and haplotype diversity of Andaman local duck

The mtDNA D-loop sequences of 71 Andaman local ducks were partially sequenced and the size of the nucleotides obtained was 657 bp after editing and trimming of primer sequences. The generated sequences were deposited to GenBank with Accession no. MK854486-MK854556. The analysis of the sequence polymorphism revealed 638 sites as monomorphic and 19 sites as polymorphic. Of the 19 polymorphic sites, one was singleton variable site and 18 were parsimony informative sites. A total number of 8 transitions and 11 transversions with transition/transversion bias (R) of 0.93 were observed. The DNA polymorphism analysis showed to have nucleotide diversity ($\pi \pm$ SD) of data sequence as 0.00897 ± 0.00078.

In Andaman Local duck, a total of 13 haplotypes (Hap A-Hap M) were detected with haplotype diversity (Hd ± SD) of 0.881 ± 0.017. The most frequent haplotype was Hap G (n = 15) followed by Hap B (n = 13), Hap I (n = 10), Hap L (n = 8), Hap F (n = 7), Hap K and M (n = 4 each) and Hap D (n = 3). Four haplotypes (Hap A, Hap C, Hap E and Hap J) were represented by a single sequence each. The number of sequences in each haplotype and the frequencies of the detected haplotypes were presented in S1 Fig. The evolutionary relationship and haplotype map of the detected haplotypes are presented in Figs 1 and 2 respectively. It was found that the haplotypes formed three clusters; Hap A, Hap C, Hap D, Hap E, Hap F in one cluster, Hap B, Hap K in another cluster and Hap G, Hap H, Hap I, Hap J, Hap L, Hap M formed a third cluster.

The sequence information of the ALD haplotypes were compared with mallard reference sequence (NC_009684) and DNA polymorphism parameters (monomorphic sites, polymorphic sites, nucleotide diversity ($\pi$) and average number of nucleotide differences (k)) were calculated (Table 1). From the analysis, it was found that Hap B had lowest number of polymorphic sites and lowest $\pi$ and k values and Hap A had highest number of polymorphic sites and highest $\pi$ and k values as compared to those of other haplotypes indicating that Hap B was comparatively closer to reference sequence. When the haplotypes of ALD were aligned with duck reference sequence (NC_009684), a total of 22 variable / polymorphic sites were detected. The multiple sequence alignment of the variable sites was presented in Fig 3.

Pair-wise nucleotide distances among different haplotypes and between different haplotypes with reference sequence are presented in Table 2. Based on the analysis, it was found that Hap B (0.00458) was closest to reference sequence followed by Hap K (0.00611) and Hap G (0.00919). Among the 13 haplotypes of ALD, the pair-wise nucleotide differences ranged from 0.00152 to 0.0248. The lowest genetic distance (0.00152) existed between Hap A and hap B,

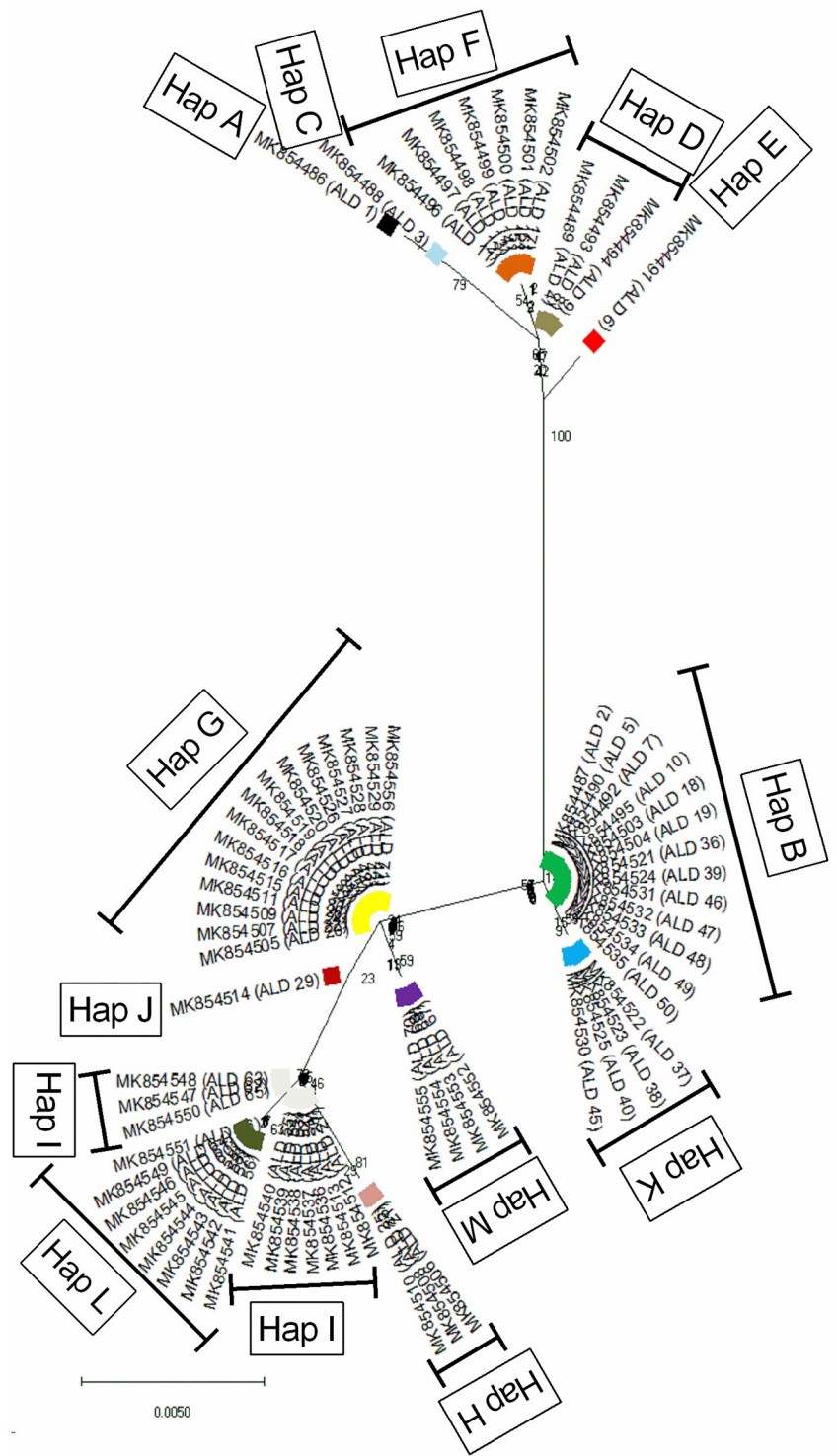

**Fig 1. Evolutionary relationship of the 13 detected haplotypes of Andaman local duck (ALD).** The evolutionary relationship was established using the Neighbor-Joining method using Tamura-Nei model as implemented in MEGAX following 1,000 bootstrap replications.

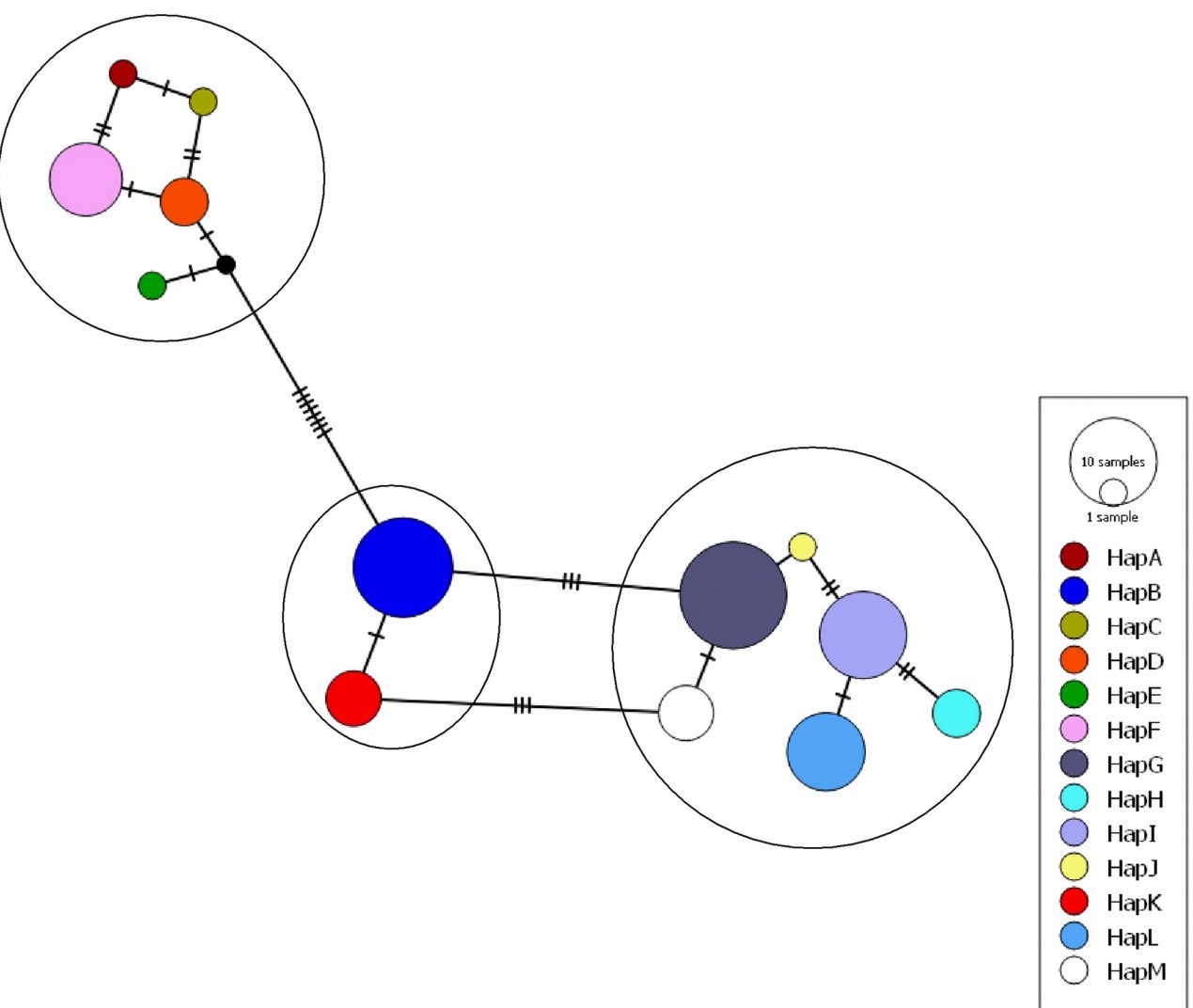

**Fig 2. Network profile of different haplotypes of ALD.** Circle area (node) is proportional to frequency. Dashed lines represent mutations. Network was drawn in PopART ver. 1.7 [39] with default settings.

Hap C and Hap K, Hap E and Hap F and Hap H and Hap J. On the other hand, highest genetic distance was found between Hap B and Hap M (0.02480).

## Andaman local duck falls in Eurasian clade of mallard mtDNA sequence

Two major clades of mallard i.e. Eurasian clade and North American clade are available throughout the world [46]. To understand in which clade ALD falls, representative mtDNA D-loop sequences including both the clades were retrieved from NCBI (S2 Table) and a phylogenetic analysis was drawn. In the phylogenetic tree, Eurasian clade and North American clade were distinct and it was evident that ALD falls in Eurasian clade (Fig 4). Further, it was found that, Hap G, Hap H, Hap I, Hap J, Hap L and Hap M were in the same cluster with Kerala, India. Hap B and Hap K were in the same cluster with Thailand, Indonesia and China. On the other hand, five haplotypes (Hap A, Hap C, Hap D, Hap E and Hap F) though close to

**Table 1. Sequence polymorphism of different haplotypes of Andaman local duck (ALD) with reference sequence (NC_009684).**

| Ref sequence compared with | Monomorphic sites | Polymorphic sites | π | k |
|---|---|---|---|---|
| Hap A | 642 | 15 | 0.01522 | 10.000 |
| Hap B | 654 | 3 | 0.00304 | 2.000 |
| Hap C | 643 | 14 | 0.01421 | 9.333 |
| Hap D | 645 | 12 | 0.01218 | 8.000 |
| Hap E | 645 | 12 | 0.01218 | 8.00 |
| Hap F | 644 | 13 | 0.01319 | 8.667 |
| Hap G | 651 | 6 | 0.00609 | 4.000 |
| Hap H | 648 | 9 | 0.00913 | 6.000 |
| Hap I | 648 | 9 | 0.00913 | 6.000 |
| Hap J | 650 | 7 | 0.00710 | 4.667 |
| Hap K | 653 | 4 | 0.00406 | 2.667 |
| Hap L | 647 | 10 | 0.01015 | 6.667 |
| Hap M | 650 | 7 | 0.00710 | 4.667 |

π: Nucleotide diversity; k: Average number of nucleotide differences

Indonesia made a separate cluster (Fig 4). In the dataset, we included domestic ducks and mallards and they were indistinguishable [32].

A Median-Joining network was constructed to understand the evolutionary relationship of ALD with different domestic ducks and mallards of the world. From the network profile, it was found that North American and Eurasian ducks formed district clades. Most of the ALD haplotypes (Hap B, Hap G, Hap H, Hap I, Hap J, Hap K, Hap L and Hap M) fall under Eurasian clade, whereas five haplotypes (Hap A, Hap C, Hap F, Hap D and Hap E) though close to Eurasian clade form a separate cluster (Fig 5).

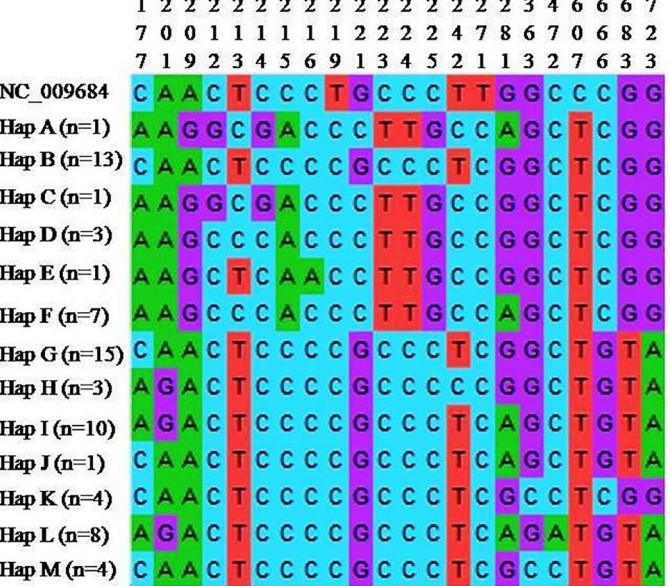

**Fig 3. Sequence alignment of different haplotypes of ALD with mallard reference sequence (NC_009684).** Only variable sites are shown. The sequence positions of every variable sites are indicated above. Multiple sequence alignment was done by ClustalW programme implemented in MEGAX.

**Table 2. Pairwise genetic distance among different haplotypes of ALD and between different haplotypes of ALD with reference sequence.**

| | NC_009684 | HapA | HapB | HapC | HapD | HapE | HapF | HapG | HapH | HapI | HapJ | HapK | HapL | HapM |
|---|---|---|---|---|---|---|---|---|---|---|---|---|---|---|
| NC_009684 | - | | | | | | | | | | | | | |
| HapA | 0.02321 | - | | | | | | | | | | | | |
| HapB | 0.00458 | 0.01851 | - | | | | | | | | | | | |
| HapC | 0.02165 | 0.00152 | 0.01696 | - | | | | | | | | | | |
| HapD | 0.01850 | 0.00458 | 0.01384 | 0.00305 | - | | | | | | | | | |
| HapE | 0.01851 | 0.00765 | 0.01384 | 0.00612 | 0.00305 | - | | | | | | | | |
| HapF | 0.02006 | 0.00305 | 0.01538 | 0.00458 | 0.00152 | 0.00458 | - | | | | | | | |
| HapG | 0.00919 | 0.02322 | 0.00458 | 0.02166 | 0.01851 | 0.01852 | 0.02007 | - | | | | | | |
| HapH | 0.01383 | 0.02163 | 0.00919 | 0.02007 | 0.01694 | 0.01695 | 0.01849 | 0.00458 | - | | | | | |
| HapI | 0.01382 | 0.02164 | 0.00919 | 0.02320 | 0.02005 | 0.02006 | 0.01850 | 0.00458 | 0.00305 | - | | | | |
| HapJ | 0.01073 | 0.02166 | 0.00611 | 0.02322 | 0.02007 | 0.02007 | 0.01851 | 0.00152 | 0.00611 | 0.00305 | - | | | |
| HapK | 0.00611 | 0.02008 | 0.00152 | 0.01853 | 0.01540 | 0.01540 | 0.01695 | 0.00612 | 0.01074 | 0.01073 | 0.00765 | - | | |
| HapL | 0.01538 | 0.02322 | 0.01073 | 0.02478 | 0.02162 | 0.02163 | 0.02007 | 0.00611 | 0.00458 | 0.00152 | 0.00458 | 0.01228 | - | |
| HapM | 0.01074 | 0.02480 | 0.00612 | 0.02324 | 0.02008 | 0.02009 | 0.02164 | 0.00152 | 0.00611 | 0.00611 | 0.00305 | 0.00458 | 0.00765 | - |

Pair-wise genetic distance was calculated using the Maximum Composite Likelihood model [37] implemented in MEGAX.

### Genetic divergence between ALD and other mallard population

Genetic divergence between ALD with Asian, European and North American mallards was calculated by DNASP. Average number of nucleotide differences, average number of nucleotide substitution per site (Dxy) and number of net nucleotide substitution per site (Da) were calculated and found lowest between ALD and Asian population followed by European population and North American population (Figs 6 and 7).

### Genetic differentiation between ALD with mallards

Genetic differentiation between ALD with Asian, European and North American clade was assessed by Wright's F-statistics (Table 3). It was found that the value was lowest when ALD was compared with Asian clade (Fst = 0.20709) and highest when ALD was compared with North American clade (Fst = 0.65821). Fst value, when ALD was compared with European (Fst = 0.22391), was very close to the value of ALD vs Asian value. An analysis of molecular variance (AMOVA) revealed that 59.62% variation lies within population and 40.38% among population (Table 4).

### Population demographic history of Andaman local duck

We calculated mismatch distribution and neutrality tests to understand the demographic history of Andaman local duck. Mismatch distribution of ALD (Fig 8) was multimodal and ragged shape in nature, indicating that population is at equilibrium and there was no recent population expansion. Neutrality test results (Tajima's D test, Fu's FS test, Fu and Li's D test, Fu and Li's F test) showed positive and non-significant values for all the tests which indicates that population is neutrally evolving.

## Discussion

Poultry farming plays a pivotal role in livelihood security of community living in rural and semi-urban areas of Southeast Asian countries [47]. Moreover, poultry is deeply involved in social and cultural life of rural people [48]. Domestic ducks are very hardy, need nominal care

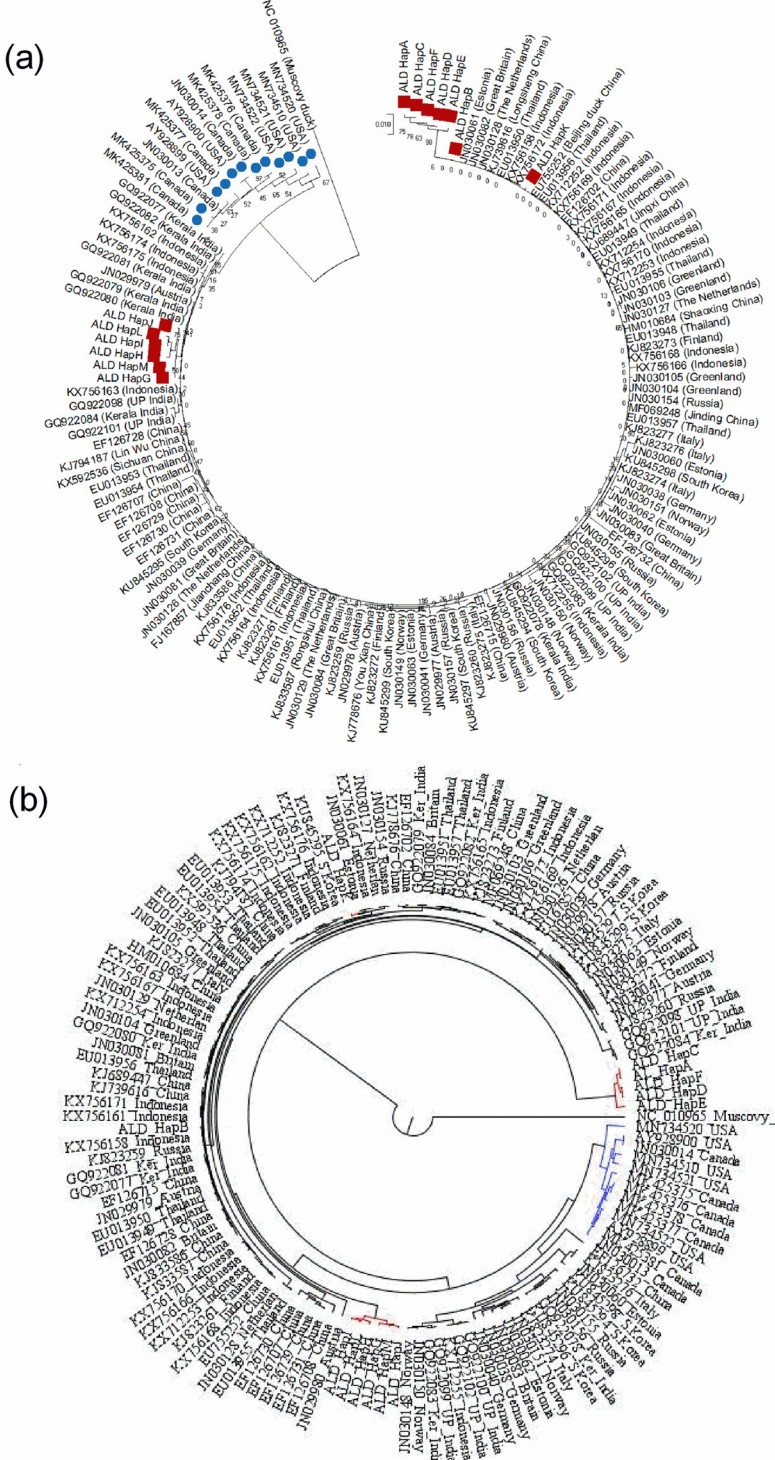

**Fig 4. Phylogenetic relationship of ALD with different domestic duck and mallard population of world representing Eurasian clade and North American clade.** (a) Neighbor-Joining based phylogenetic tree, (b) Bayesian phylogenetic tree. In N-J phylogenetic tree, sequences representing 'North American clade' and ALD are presented with blue circles and brown squares respectively. In Bayesian phylogenetic tree, sequences representing 'North American clade' and ALD are presented with blue and brown lines respectively. The Neighbor-Joining tree was constructed using Tamura-Nei model as implemented in MEGAX following 1,000 bootstrap replications. The Bayesian phylogenetic tree was drawn in BEAST v1.10.4. Muscovy duck (NC_010965) was used as an outgroup.

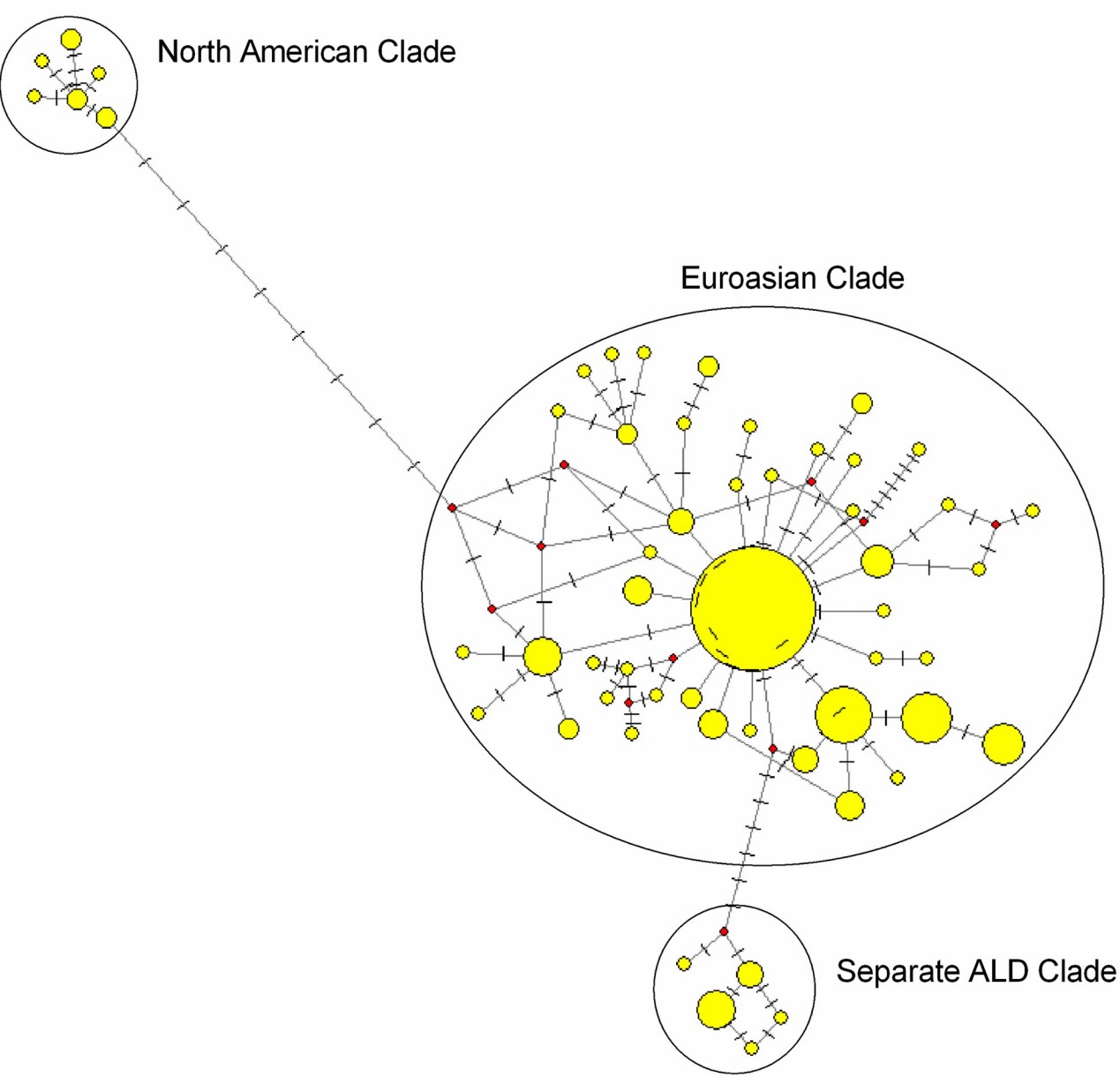

**Fig 5. Network profile of the mtDNA control region sequences in ALD and mallards representing Eurasian and North American clade.** Circle area (node) is proportional to frequency. Dashed lines represent mutations. Network was drawn in Network 10 [38] with default settings.

and management and can sustain in almost any type of environmental conditions [49]. Therefore, duck raising is extremely popular in India especially in Eastern, North Eastern and Southern parts. Most of the ducks reared in India is indigenous or desi type and possess specific distinct characters. Conservation of these indigenous breeds is very crucial for maintenance of biodiversity and ecosystem functions and effective conservation of any breed rely upon proper recognition through genomic characterization [50]. Andaman local duck, found in a small isolated pocket of Middle and North Andaman, is a native duck genetic resources and on the edge of extinction due to negligence and pressure from exotic duck breeds seeking immediate attention and conservation efforts. Information on genetic origin, population structure and genetic diversity is required for formulation of breeding plan and conservation strategy of

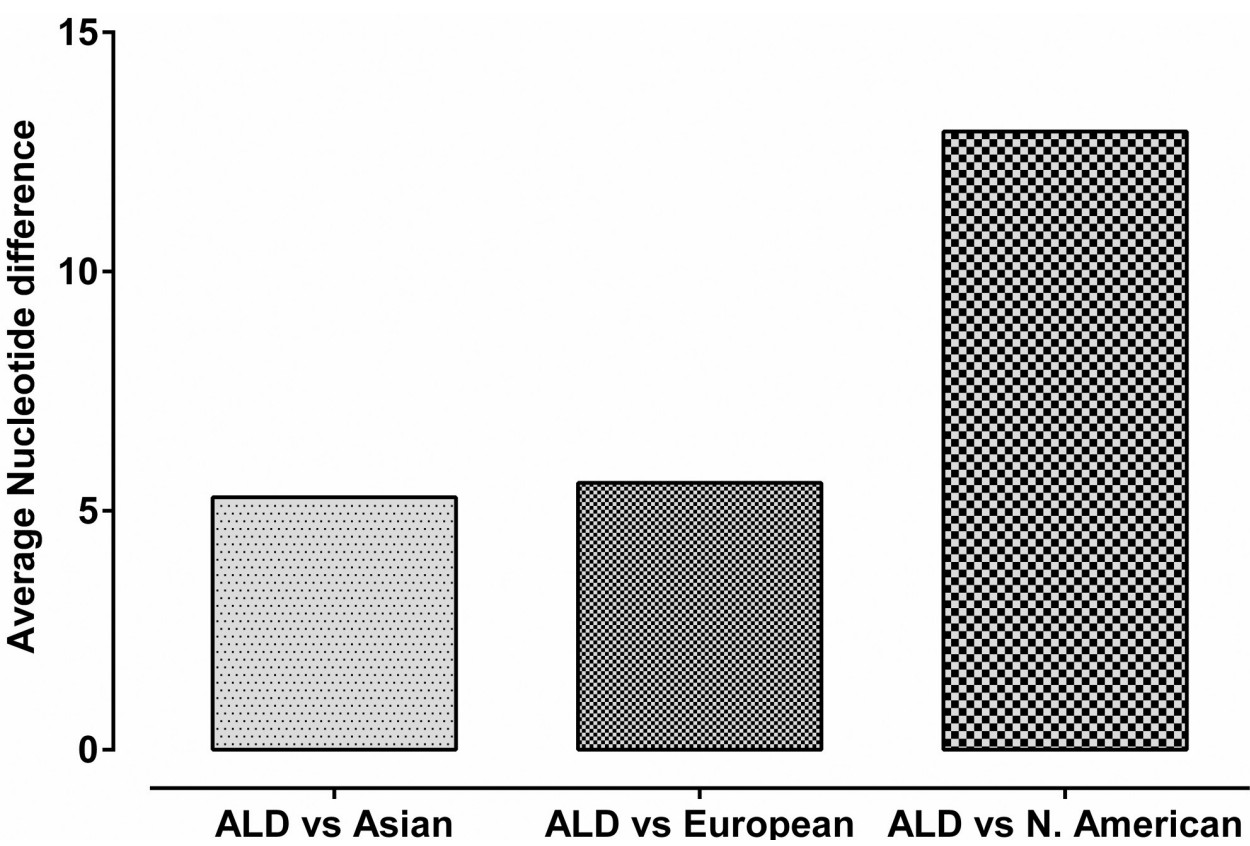

**Fig 6. Average number of nucleotide differences between ALD with Asian, European and North American mallards.**

this threatened breed [33]. In the present study, the genetic diversity, population genetic parameters, phytogeography, genetic origin and migration route of the duck breed has been investigated.

In the present study, 19 polymorphic sites with nucleotide diversity ($\pi \pm$ SD) of 0.00897 $\pm$ 0.00078 were detected in ALD sequences. Total 13 haplotypes were detected with haplotype diversity (Hd $\pm$ SD) of 0.881 $\pm$ 0.017. The nucleotide diversity of ALD was higher than those of Chinese mallards (0.00202), Eurasian mallards (0.00600), Russian mallards (0.00600), Chinese spot billed ducks (0.00425), USA mottled ducks (0.00198) [33], ducks of Western Russia (0.0052), North Asia (0.0083) [28], Chinese domestic duck (0.00218) [32], Bangladeshi duck (0.00056–0.00372) [51], and Korean duck population (0.00195–0.00372) [51] but was lower than USA mallards (0.01668) [33] and Alaskan ducks (0.0130) [28]. On the other hand, the overall haplotype diversity in our study population was closer to those observed for Eurasian mallards (0.843) [33], Chinese spot billed ducks (0.8301) [33], Bangladeshi ducks (0.337–0.886) [51] and Thai native duck (0.700–0.800) [52], was lower than Russian mallards (0.957), USA mallards (0.933) [33], Northasian ducks (0.9872) and Alaskan ducks (0.9806) [28] and was higher than Chinese mallards (0.472) and USA mottled ducks (0.7500) [33]. The high genetic diversity of ALD indicates introgression of genetic material from other breeds. The reason might be that ANI are relatively remote and ducks are generally reared in free range system with high chance of mixing with ducks of other indigenous breeds. Intensive and directional breeding for a particular trait reduces genetic diversity and sometimes even leads to loss of biodiversity [53, 54]. In case of ALD, no selective breeding has been

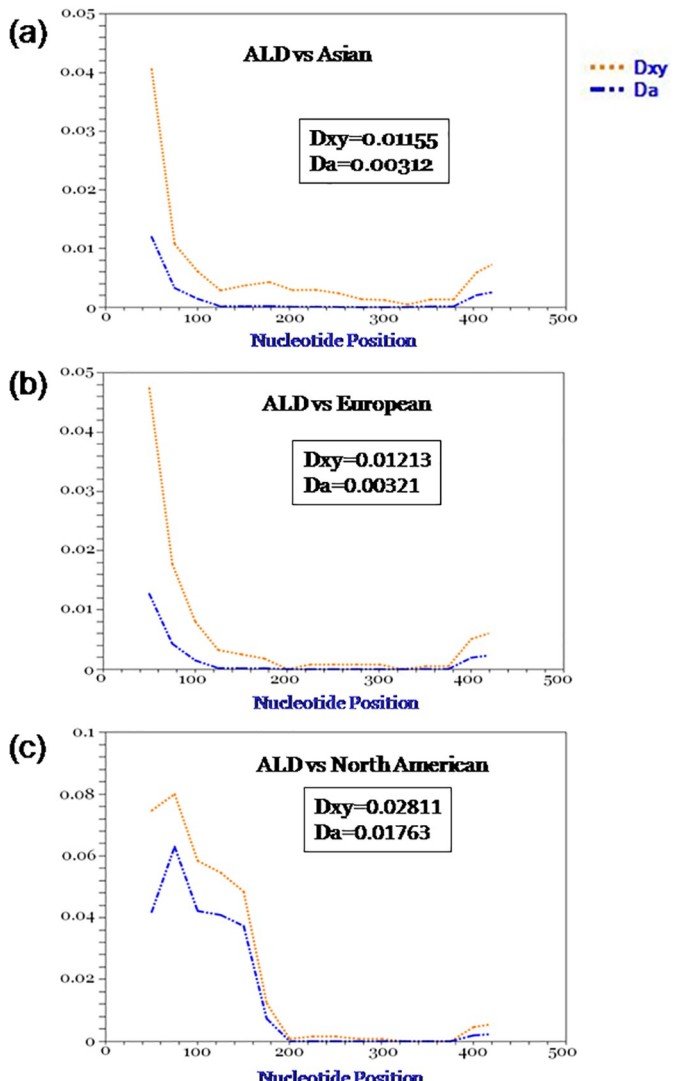

**Fig 7. Genetic divergence between ALD and other mallard population.** Dxy denotes average number of nucleotide substitution per site and Da denotes number of net nucleotide substitution per site.

practiced and mating happens naturally in the environment; this probably might be the cause behind high genetic diversity of the population. Moreover, it can be postulated that ALD bearing high genetic diversity has strong ability to adapt to environmental changes [55, 56] as genetic diversity is positively correlated with the capability to adapt to environmental changes

**Table 3. Pairwise *F*st values of ALD and other major duck clades.**

|  | 1 | 2 | 3 | 4 |
|---|---|---|---|---|
| 1. ALD | - |  |  |  |
| 2. European | 0.22391* | - |  |  |
| 3. North American | 0.65821* | 0.73735* | - |  |
| 4. Asian | 0.20709* | 0.05641 | 0.79766* | - |

Fst values marked by an asterisk are statistically significant.

**Table 4. AMOVA analysis of ALD and different mallard clades.**

|  | d.f. | SS | VC | % var |
|---|---|---|---|---|
| Among population | 3 | 122.074 | 0.87665 | 40.38 |
| Within population | 191 | 247.223 | 1.29436 | 59.62 |
| Total | 194 | 369.297 | 2.17102 | - |

All variances are statistically significant (p<0.05). Abbreviations: d.f.: degree of freedom; SS: sum of squares; VC: Variance components; % var: percentage of variation

[57]. As this local breed has strong ability of adapt to environmental changes, it is highly probable that it can withstand impending climate change. Therefore, conservation of this breed is extremely essential.

Animal domestication in Neolithic age is a landmark and evolutionary event in history of human civilization as it transformed human from hunting to agriculture based occupation [58]. Southeast Asian countries are considered as an epicenter of animal domestication. Mitochondrial DNA based analysis indicated presence of two broad clades of mallards; clade A of Asia and Clade B of North-America [28, 59]. In another study, Hou et al. [33], found that Eurasian domestic ducks and mallards were distinct from North American mallards. Finally, Kraus et al. [46], investigated the population genetic structure of mallards from Asia, Europe

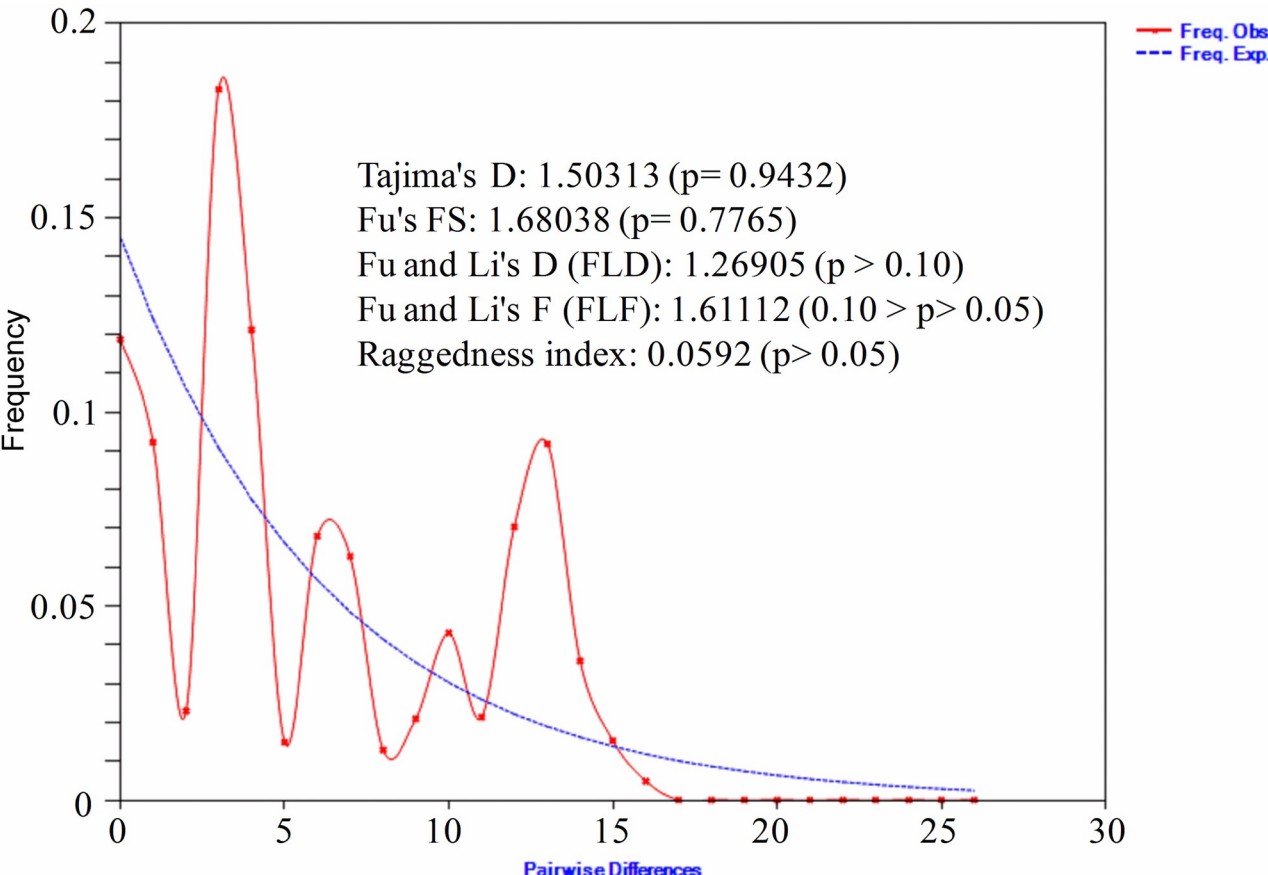

**Fig 8. Mismatch distribution graph and neutrality tests of Andaman local ducks.** The x and y axis presents number of pairwise differences and the frequency of the pairwise comparisons respectively.

and America and redefined the two clades as 'Euroasian clade' and 'North American clade'. In the present study, it was found that the ALD belonged to 'Euroasian clade' and three major clusters were observed (Fig 4); one cluster was phylogenetically close to Southeast Asian countries, another to mainland India and a third one formed independent cluster indicating that they evolved locally.

Andaman archipelago lie in Bay of Bengal in a curve between Burma and Indonesia. Its close proximity to Southeast Asian countries prompted cultural linkage and material exchange since time immemorial [60]. Trade, cultural and political relation were established between kingdoms of Indian subcontinent and Southeast Asian kingdoms in Burma, Thailand, Indonesia, Philippines, Cambodia and Champa starting around 290 BC and maritime trade exchanges were via Andaman and Nicobar islands, which are situated in the sea route between Indian and Southeast Asia, and regarded as one of the finest natural harbours [61, 62]. Moreover, the mitogenome based analysis revealed the origin of Andaman islanders from Southeast Asian countries (Malaysia, Thailand) and China [63–65]. Migration of indigenous people must have been accompanied by livestock and poultry resources. In addition, the colonial people who ruled Andaman and Nicobar islands, in different phases introduced several livestock and poultry breeds from Southeast Asian countries for their consumption purpose [66, 67]. These breeds over the time evolved in the isolated places and act as the founding population for the indigenous livestock and poultry breeds of ANI. Recently, mitochondrial marker based tracing the genetic root of Trinket cattle established this theory [68]. Therefore, presence of Southeast Asian gene pool in these duck haplotypes might be the outcome of commercial exchange either by Andaman indigenes or by people of colonial era. In post-independence, people from different parts of India especially Southern parts were settled in North and Middle Andaman. This is highly probable that they brought poultry species along with them as it is amenable to carry. This migration of human from Southern parts of mainland India explains the presence of South Indian haplotypes in ALD.

Independent ALD cluster indicates that it has evolved in the microenvironment. New breed or haplotype evolves in a particular area either due to of artificial selection by humans or natural selection by the environment [69]. In our case, artificial selection is highly improbable as there is no selective or directional breeding; therefore natural selection pressure imposed by environmental conditions, climate, parasites and predators might be the driving force behind evolution of the novel haplotypes. This makes sense as these islands are very isolated with a typical microclimatic condition. Charles Darwin [70] in his landmark book "On the Origin of Species" mentioned that "adaptive evolution occurs in the wild by natural selection"; Darwin's theory supports our hypothesis. To fully understand local adaptation of ALD, genome level variation using neutral or non-neutral markers need to be explored in future study.

Genetic divergence of ALD from rest of the mallard population as shown in Figs 6 and 7 might be due to accumulation of mutations as a result of reproductive isolation of ALD [71]. As ANI is geographically isolated from rest of the world by sea, some novel adaptation via natural selection or as a result of genetic drift led to the genetic differentiation in ALD. Restricted gene flow due to geographical isolation and island microenvironment/habitat of ALD led to genetic difference from central mallard population [72]. Another possible explanation of the genetic divergence might be the founder effect [73] in which certain population acquired some gene mutation due to selection pressure and acted as the founding population for ALD.

The demographic history of ALD as inferred from mismatch distribution indicates that the population is at demographic equilibrium [74]. Moreover, the ragged shape of the mismatch distribution is indicative of widespread population lineage [75]. The data was good fit to a population expansion model as the raggedness index of the population (Fig 8) was found non-significant [76]. Positive and non-significant values of the neutrality tests support the neutral

theory of molecular evolution [77] and evolutionary changes in Andaman local ducks arose due to random genetic drifts. This is in agreement with the hypothesis of natural selection postulated by Charles Darwin.

This is the first report on genetic characterization of Andaman local duck. This may be concluded that Andaman local duck migrated from Southeast Asian countries and Southern parts of mainland India along with the migration of indigenes, colonial people or settlers. Moreover, some haplotypes of Andaman local duck evolved locally due to natural selection pressure.

## Supporting information

**S1 Table. Details of sampling location of Andaman local duck.**
(DOCX)

**S2 Table. Details of the sequences used in the present study.**
(DOCX)

**S1 Fig. The number of sequences in each haplotype and the frequencies of the detected haplotypes of Andaman local duck.**
(TIF)

## Author Contributions

**Conceptualization:** Arun Kumar De.

**Data curation:** Sneha Sawhney, Perumal Ponraj.

**Formal analysis:** Arun Kumar De, Debasis Bhattacharya, Dhruba Malakar.

**Funding acquisition:** Arun Kumar De.

**Investigation:** Arun Kumar De, A. Kundu.

**Methodology:** Arun Kumar De, Samiran Mondal, Dhruba Malakar.

**Resources:** T. Sujatha, Jai Sunder.

**Software:** S. K. Ravi.

**Writing – original draft:** Arun Kumar De.

**Writing – review & editing:** Debasis Bhattacharya.

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
