## [Decision Letter · Decision Letter 0]

7 Oct 2020

PONE-D-20-16012

Origin, genetic diversity and evolution of Andaman Local Duck, a native duck germplasm of an insular region of India

PLOS ONE

Dear Dr. De,

Thank you for submitting your manuscript to PLOS ONE. After careful consideration, we feel that it has merit but does not fully meet PLOS ONE’s publication criteria as it currently stands. Therefore, we invite you to submit a revised version of the manuscript that addresses the points raised during the review process.

The study idea has merits and an appropriate revision would be very useful bring the novelty. Therefore, a point by point reply to the questions of reviewers will be highly appreciated.

We look forward to receiving your revised manuscript.

Kind regards,

Gyaneshwer Chaubey, Ph.D.

Academic Editor

PLOS ONE

Journal Requirements:

2. In your Methods section, please provide additional location information, including geographic coordinates for the data set if available.

"Authors are thankful to NABARD and Department of Biotechnology, Ministry of Science and Technology, Government of India (grant no. BT/BI/04/066/2004) for providing fund."

"No"

Reviewers' comments:

Reviewer's Responses to Questions

**Comments to the Author**

1. Is the manuscript technically sound, and do the data support the conclusions?

Reviewer #1: Partly

Reviewer #2: Yes

2. Has the statistical analysis been performed appropriately and rigorously? 

Reviewer #1: No

Reviewer #2: Yes

3. Have the authors made all data underlying the findings in their manuscript fully available?

Reviewer #1: No

Reviewer #2: Yes

4. Is the manuscript presented in an intelligible fashion and written in standard English?

Reviewer #1: Yes

Reviewer #2: Yes

5. Review Comments to the Author

Reviewer #1: Comments to the Author

The manuscript entitled “Origin, genetic diversity and evolution of Andaman local duck, a native duck germplasm of an insular region of India” has been reviewed and sequences from the partial mt-DNA D-loop region were identified for the 71 Andaman local ducks and haplotype diversity was carried out by comparing existing haplotypes in GenBank. Authors concluded Andaman duck haplogroup belongs to the Eurasian clade that includes South east Asia as the main root of origin. In some reasons, this manuscript is useful for the conservation of the local duck species in India. However, in my point of view, the publication in the PLoS One Journal is not appropriate and it’s better to submit another Journal, focusing on the conservations of exotic species.

Major comments:

(1) The selected 71 Andaman local ducks are not sufficient for the phylogenetic analysis and it’s better to increase the number of samples. Also, details for sampling locations is necessary, you can include a supplementary figure for sample distribution of ALD (approximately 13 villages).

(2) It’s not relevant that authors conclude the local adaptation and resilient of ALD by sole evaluation of the D-loop. We must be considered genome level variation by using neutral or non-neutral markers that necessarily described the local adaptation, selection signature in this breed. if they have highly adapted to local condition in Andaman and Nicobar Island and happened to undergone natural selection it must be clear from their adaptive genetic variations also related fitness traits etc.

(3) Line 305-309, what are evidence for having high adaptation of these breeds to climatic change?

(4) Some of the paragraphs (310-333, & 334-361) in the discussion are more elaborate the origin of haplogroups, which are too long, delete less informative facts and rearrange.

(5) Its better if you could include the proportion of each breed within each haplogroup. In Fig 2, it is not clear what each color are represented for? Revise the Fig 2 with a key for each color code.

(6) I don’t think you need so much reiteration of the data in the results and discussion sections. Especially author repeats the results of haplo groups belongs to each cluster. Just give table number, the reader can look at the tables if they wish more details.

(7) No discussion is included regarding the divergence estimations between ALD and other mallard population (Fig. 6, Fig. 7). Authors need to emphasize the relevance of these results with natural selection for maintaining the genetic diversity in ADL.

(8) Did you consider any evidence for ancients’ population expansion for these populations?

Reviewer #2: Taking in a view of an endangered species it can be a good effort to understand genetic structure and relatives of species to save them from being extinct completely. After getting the genetic makeup of a species it will be helpful in reintroducing them in environment.

6. PLOS authors have the option to publish the peer review history of their article (what does this mean?). If published, this will include your full peer review and any attached files.

Reviewer #1: No

Reviewer #2: No

---

## [Author Response · Author response to Decision Letter 0]

9 Dec 2020

Editor's Comments

Comment: 1. Please ensure that your manuscript meets PLOS ONE's style requirements, including those for file naming. The PLOS ONE style templates can be found at

Reply: The manuscript meets the PLOS ONE's style requirements.

Comment: 2. In your Methods section, please provide additional location information, including geographic coordinates for the data set if available.

Reply: Location information including geographic coordinates of the samples has been included as Table S1.

Comment: 3. Thank you for stating the following in the Acknowledgments Section of your manuscript:

"Authors are thankful to NABARD and Department of Biotechnology, Ministry of Science and Technology, Government of India (grant no. BT/BI/04/066/2004) for providing fund."

"No"

Reply: Necessary modification has been made in revised manuscript. The funding statement should be as follows;

The work was funded by National Bank for Agriculture and Rural Development (NABARD, Port Blair) and Department of Biotechnology, Ministry of Science and Technology, Government of India (grant no. BT/BI/04/066/2004). 

I have included this in covering letter.

Comment: 4. We note that you have stated that you will provide repository information for your data at acceptance. Should your manuscript be accepted for publication, we will hold it until you provide the relevant accession numbers or DOIs necessary to access your data. If you wish to make changes to your Data Availability statement, please describe these changes in your cover letter and we will update your Data Availability statement to reflect the information you provide.

Reply: Modifications has been made.

Comment: 5. Please include captions for your Supporting Information files at the end of your manuscript, and update any in-text citations to match accordingly. Please see our Supporting Information guidelines for more information: http://journals.plos.org/plosone/s/supporting-information.

Reply: Included in the revised manuscript.

Reviewer #1: Comments to the Author

Comment: The manuscript entitled “Origin, genetic diversity and evolution of Andaman local duck, a native duck germplasm of an insular region of India” has been reviewed and sequences from the partial mt-DNA D-loop region were identified for the 71 Andaman local ducks and haplotype diversity was carried out by comparing existing haplotypes in GenBank. Authors concluded Andaman duck haplogroup belongs to the Eurasian clade that includes South east Asia as the main root of origin. In some reasons, this manuscript is useful for the conservation of the local duck species in India. However, in my point of view, the publication in the PLoS One Journal is not appropriate and it’s better to submit another Journal, focusing on the conservations of exotic species.

Reply: The manuscript has been revised as suggested by the reviewer.

Major comments:

Comment: (1) The selected 71 Andaman local ducks are not sufficient for the phylogenetic analysis and it’s better to increase the number of samples. Also, details for sampling locations is necessary, you can include a supplementary figure for sample distribution of ALD (approximately 13 villages).

Reply: FAO guidelines (FAO, 2011) on 'sampling pattern' for genetic characterization of a breed states that at least 40 genetically unrelated animals/samples should be sampled and sampling should cover the different agroclimatic zones where the breed is found. Moreover, typically no more than 10 percent of any one herd or village population should be sampled and in any case no more than five animals should sampled from any herd. The population of Andaman duck is 42371 as per 19th Livestock census (19th Livestock Census-2012, All India Report, Ministry of Agriculture Department of Animal Husbandry, Dairying and Fisheries, Krishi Bhawan, New Delhi, accessed from https://dahd.nic.in/). Considering the small population of Andaman Local Duck, the sample size 71 is quiet reasonable. Moreover, we collected representative samples from all the geographical region where the breed is available and followed sampling pattern as recommended by FAO guidelines. We sampled not more that 3 genetically unrelated samples per village. The details of sampling location has been included as a supplementary Table (Table S1) in the revised manuscript.

Comment: (2) It’s not relevant that authors conclude the local adaptation and resilient of ALD by sole evaluation of the D-loop. We must be considered genome level variation by using neutral or non-neutral markers that necessarily described the local adaptation, selection signature in this breed. if they have highly adapted to local condition in Andaman and Nicobar Island and happened to undergone natural selection it must be clear from their adaptive genetic variations also related fitness traits etc.

Reply: The authors agree with the reviewer. As per suggestion, necessary modification has been made in the revised manuscript.

Comment: (3) Line 305-309, what are evidence for having high adaptation of these breeds to climatic change?

Reply: In an evolutionary biology context, adaptation is the ability of a breed to adjust to a given environment. Farmers are rearing Andaman local duck from a long time. There has been extensive climate change in Andaman and Nicobar islands over the last two decades. Moreover, after Tsunami in 2004, a sea change in climatic condition in these islands has been observed. The indigenous duck population has maintained their production potential and population status irrespective of the climate change whereas other breeds of ducks has either became extinct or population has reduced drastically. Therefore, it is reasonable to assume that they are highly adaptable to the local condition. 

Comment: (4) Some of the paragraphs (310-333, & 334-361) in the discussion are more elaborate the origin of haplogroups, which are too long, delete less informative facts and rearrange.

Reply: As suggested, necessary modification has been made and included in the revised manuscript.

Comment: (5) Its better if you could include the proportion of each breed within each haplogroup. In Fig 2, it is not clear what each color are represented for? Revise the Fig 2 with a key for each color code.

Reply: As suggested, necessary modification has been made and included in the revised manuscript.

Comment: (6) I don’t think you need so much reiteration of the data in the results and discussion sections. Especially author repeats the results of haplo groups belongs to each cluster. Just give table number, the reader can look at the tables if they wish more details.

Reply: As suggested, necessary modification has been made and included in the revised manuscript.

Comment: (7) No discussion is included regarding the divergence estimations between ALD and other mallard population (Fig. 6, Fig. 7). Authors need to emphasize the relevance of these results with natural selection for maintaining the genetic diversity in ADL.

Reply: A paragraph on this has been incorporated in the discussion section.

Comment: (8) Did you consider any evidence for ancients’ population expansion for these populations?

Reply: Population demographic history of Andaman Local Duck based on mismatch distribution and neutrality tests has been included in the revised manuscript.

Comment: Reviewer #2: Taking in a view of an endangered species it can be a good effort to understand genetic structure and relatives of species to save them from being extinct completely. After getting the genetic makeup of a species it will be helpful in reintroducing them in environment.

Reply: The authors are thankful to the reviewer for positive response.

---

## [Editor Report · Decision Letter 1]

23 Dec 2020

Origin, genetic diversity and evolution of Andaman Local Duck, a native duck germplasm of an insular region of India

PONE-D-20-16012R1

Dear Dr. De,

We’re pleased to inform you that your manuscript has been judged scientifically suitable for publication and will be formally accepted for publication once it meets all outstanding technical requirements.

Kind regards,

Gyaneshwer Chaubey, Ph.D.

Academic Editor

PLOS ONE
---

## [Editor Report · Acceptance letter]

29 Jan 2021

PONE-D-20-16012R1 

Origin, genetic diversity and evolution of Andaman Local Duck, a native duck germplasm of an insular region of India 

Dear Dr. De:

I'm pleased to inform you that your manuscript has been deemed suitable for publication in PLOS ONE. Congratulations! Your manuscript is now with our production department. 

Kind regards, 

on behalf of

Dr. Gyaneshwer Chaubey 

Academic Editor

PLOS ONE